# Sex differences in health care expenditures and mortality after spousal bereavement: A register-based Danish cohort study

**Alexandros Katsiferis**[1,2]*, **Samir Bhatt**[1,3], **Laust Hvas Mortensen**[1,2], **Swapnil Mishra**[1,4], **Rudi G. J. Westendorp**[1,2]

**1** Section for Epidemiology, Department of Public Health, University of Copenhagen, Copenhagen, Denmark, **2** Statistics Denmark, Copenhagen, Denmark, **3** Department of Infectious Disease Epidemiology, Imperial College London, London, United Kingdom, **4** Saw Swee Hock School of Public Health, National University of Singapore, Singapore, Singapore

\* alexandros.katsiferis@sund.ku.dk

**Data Availability Statement:** The current study uses detailed data on individuals. This means that even if direct identifiers like name, date of birth and street address are removed from the data, it is still

## Abstract

### Background

Spousal bereavement is a life event that affects older people differently. We investigated the impact of spousal bereavement on medical expenditures and mortality in the general population, emphasizing on age and sex.

### Methods

Data are from a population-based, retrospective cohort study following 924,958 Danish citizens over the age of 65 years, within 2011–2016. Changes in health care expenditures in those who suffer bereavement were compared with time matched changes among those who did not. Mortality hazards were analysed with time to event analysis.

### Results

A total of 77,722 (~8.4%) individuals experienced bereavement, 65.8% being females. Among males, bereavement was associated with increase of expenditures the year after, that was 42 Euros per week (95% CI, 36 to 48) larger than the non-bereaved group. The corresponding increase for females was 35 Euros per week (95% CI, 30 to 40). The increase of mortality hazards was highest in the first year after bereavement, higher in males than females, in young old and almost absent in the oldest old. Compared with the reference, mortality the year after spousal loss was 70% higher (HR 1.70 [95% CI 1.40 to 2.08]) for males aged 65–69 years and remained elevated for a period of six years. Mortality for females aged 65–69 years was 27% higher in the first year (HR 1.27, [1.07 to 1.52]), normalizing thereafter.

### Conclusion

Bereavement affects older people differently with younger males being most frail with limited recovery potential.

possible to re-identify the individuals in the study, which would breach basic principles of data protection. Consequently, the data can only be shared under specific conditions. According to Danish law, scientific organizations can be authorized to work with data within Statistics Denmark and can provide access to individual scientists inside and outside of Denmark. Data are available via the Research Service Department at Statistics Denmark: (www.dst.dk/da/TilSalg/Forskningsservice) for researchers who meet the criteria for access to confidential data.

**Funding:** AK, RGJW, and LHM are supported by The Novo Nordisk Foundation Challenge Programme for the project Harnessing the Power of Big Data to Address the Societal Challenge of Aging (NNF17OC0027812). SB acknowledges support from The Novo Nordisk Foundation via the Novo Nordisk Young Investigator Award (NNF20OC0059309), which also supports SM. LHM is employed at Statistics Denmark, the national Danish Statistics office, which organization holds the right to the registry data used in the study. The funders had no role in the study design, data collection and analysis, decision to publish, or preparation of the manuscript. There was no additional external funding received for this study.

**Competing interests:** I have read the journal's policy and the authors of this manuscript have the following competing interests: LHM is employed at Statistics Denmark, the national Danish Statistics office. This does not alter our adherence to PLOS ONE policies on sharing data and materials.

## Introduction

Major life events such as bereavement are stressors that effect individual's behavior, well-being, and health [1]. The ability of individuals to maintain function when perturbed by these types of stressors is defined as resilience [2]. In medicine the term is of particular interest since it allows to predict whether a patient can withstand future health-shocks and or the accompanying interventions. The need for tools that measure and provide insights regarding resilience in older adults is thus one of the top priorities in the field of aging [3–9]. The process of ageing inevitably comes with incremental damage to molecules, tissues and organs which makes people increasingly vulnerable to stressors, a process named frailty, describing the body's inability to maintain homeostasis and efficiently regulate all those physiological, systemic variables which account for their respective recovery potential [10]. However, resilience, also referred to adaptive capacity or recovery potential, is a heterogeneous phenomenon, given the fact that every individual is exposed to a unique series of stressors over their life span [11]. It is also a dynamic trait interacting with age and gender as the ageing process affecting the two sexes differently [12].

Spousal bereavement, i.e., death of a spouse, is a major life event linked with various mental, physical, and behavioral complications and also associated with increased hospitalization and mortality hazard [13–17], known as the 'widowhood effect' [18]. Importantly, the effect of bereavement on mortality appears to be modified by age and sex, and time since the spouse passed away [19–24]. The socioeconomic status of individuals, the amount of (in)formal care, as well as the pre-existing medical condition are factors that influence the widowhood effect and provide targets for intervention, especially for older adults [22, 23]. The changes in health care use of spousal bereaved individuals have also been studied, indicating a sex and age effect. However, data on health care use were limited to medications and primary care annual visits, thus not considering the use of home and residential care, the latter being highly pertinent for older adults [25]. In general, spousal loss may be associated with a variety of different health related complications preceding bereavement and these parallel changes have been poorly studied.

We investigated and compared the health status of bereaved individuals both before and after spousal loss with non-bereaved ones using overall health care expenditures derived from unique longitudinal, Danish national registries. The rationale behind using these expenditures is that they serve as a quantitative indicator or proxy of a person's health status [26]. The changes in health expenditures were then related with patterns of all-cause mortality in an attempt to detect sex differences and age differentials in the consequences of bereavement.

## Methods

### Data processing

The data analyzed are available in the Danish nationwide registers being kept at Statistics Denmark (https://www.dst.dk) providing a variety of information at baseline, such as socio-demographic variables (sex, date of birth, date of death, number of comorbidities, affluence index which combines income and savings at the start of the cohort categorized in percentiles [27, 28], and number of children), as well as the date of spousal bereavement, which is the variable of interest. Furthermore, we also monitored health care expenditures of Danish citizens, who had been resident in Denmark for at least 5 years before their date of death, from January 1$^{st}$, 2011 to December 31$^{st}$, 2016 [27, 29]. Health care expenditures (measured in Euros), were computed for weekly intervals, grouped by different types of costs. Specifically, information on care- (home and residential care) and cure-related costs (hospitalization, prescription drug and primary care) was

available. The register sources of the data, as well as the methods used to compute the expenditures are described elsewhere [27, 30]. In the current research, we emphasized analyses of overall expenditures (the sum of care and cure costs), also showing distinct patterns in cure but also in care expenditures, which is most relevant in older age [31]. There were no missing data in the aforestated variables. Importantly, as Denmark's healthcare system is mostly tax-based funded, the available information in the registers accounts for 97% of the country's medical spending (expenditures) [29].

## Study population

The current register-based study consists of a cohort of married individuals at age of 65 or above for whom health care expenditures were monitored in the period of 2011 through 2016. Each person was followed for a maximum period of 6 years in the registers from January 1st 2011 until the date of death, migration or censoring on the December 31st 2016.

## Health care expenditures across bereavement groups

Weekly health care expenditures were monitored two years before and one year after spousal bereavement. Hence, for the analysis of health care expenditure data, only individuals who experienced bereavement at 2013–2015 were included. In order to create a control group, we performed 1:1 nearest neighbor (NN) matching, using a logistic regression propensity score predicting the probability of being bereaved [32]. The comparison groups were constructed to be similar according to a set of matching covariates used to extract the scores, the latter being age at entry, sex, affluence index as a measure of socioeconomic status, number of comorbidities, and number of children as a proxy of informal caregiving.

## Statistical analysis

We analyzed the health care expenditure data via the difference in differences (DiD) approach, using linear regression models, to study the differential effect of bereavement on the treatment (spousal bereaved) group versus the control (non-bereaved) group [33]. The outcome variable of the regression models was the averaged, weekly health care expenditures, with the independent variables being the bereavement indicator, membership to the bereaved group or not, along with indicator variables for membership in the post-bereavement period. To obtain the average bereavement effect, we considered the interaction (DiD coefficient) between the bereavement and time indicator. The suitability of the parallel trends assumption (differences between bereaved and non-bereaved in the absence of bereavement) was assessed by visual inspection of the time series plots. We further evaluated the ability of the linear models to fit the observed data via the coefficient of determination (R-squared).

In order to assess the association of spousal loss with mortality from all causes, bereaved individuals were followed from the date of spousal death, with non-bereaved ones being followed from the date of inclusion, serving as the control group. Individuals of both groups were followed until their date of death or censoring, and there was no loss to follow-up. Time to event analysis was performed using flexible (spline-based) survival models [34], stratified for different sexes and age groups to unravel possible effect modifications. In our models, we adjusted for affluence index, as well as for the number of children and comorbidities a person has at baseline, since the association of bereavement with mortality risk could be distorted by the socioeconomic status, access to (in)formal care and number of pre-existing chronic diseases. The age groups (in 5-year intervals) were constructed based on age at the index date, i.e., a constructed variable being the age at bereavement for the bereaved group and age at the

study entry (2011-01-01) for the non-bereaved one. A description of the two study designs, DiD and mortality analysis, is illustrated in S1 Table.

The time since spousal loss was also considered for mortality analysis, since it is expected that bereavement might have a different effect in the short- compared with the long-term. Specifically, the time varying effect of bereavement was modelled by choosing a baseline natural spline smoother of the log (time) variable with four degrees of freedom (df = 4) with an interaction between the bereavement covariate and a natural spline smoother of log (time) with 4 degrees of freedom as well. The degrees of freedom were based on the smallest extracted AIC values of the models fitted [35]. The spline-based survival analysis enabled a continuous visualization of the hazard ratio through time without stratifying the analysis in time-intervals for which the proportionality assumption might not hold and even the average hazard ratio is not representative of the underlying phenomenon. Computations were performed using the R statistical language (version 4.1.2, R core team, 2022).

### Ethics statement

All methods were performed in accordance with the relevant guidelines and regulations. Danish legislation allows for register-based research of this type to be conducted without the consent of participants and without ethical committee approval. The study was conducted according to the rules of the Danish Data Protection Agency. All data was held at Statistics Denmark, which is the Danish national statistical institution.

## Results

### Population statistics

The overall study population consisted of 924,958 Danish residents, with 413,163 (44.6%) being males and 511,795 (55.4%) females. The mean age (standard deviation) at the date of inclusion was 74.3 (7.5) years with 73.3 (6.9) and 75.1 (7.9) being the descriptives for males and females, respectively. Further information on the sociodemographic variables of the study sample is summarized in Table 1. Out of this population, 77,722 (8.4%) experienced spousal bereavement within the period of 2011–2016, with 26,577 (6.4%) and 51,145 (10%) being the respective numbers for males and females. The mean age at spousal bereavement was 77.9 (6.6) years, with 78.9 (7.1) and 77.4 (6.4) depicting the means for males and females respectively.

### Bereavement and medical spending

Health care expenditures of both bereaved and non-bereaved individuals were monitored for a total of three years. In the case of bereaved persons, the follow-up period consisted of two years before the bereavement date and one year after, thus excluding those who experienced the event in 2011–2012 or 2016. Therefore, the sample size of bereaved individuals for the medical spending analysis was approximately 49% of the total bereaved sample, restricted to 38,027 individuals. Since we matched every case with a non-bereaved individual based on sex, age, affluence index, number of children and comorbidities, the sample size was equal in both groups.

The analysis of expenditures, measured in Euros and stratified in type of health care expenditures and sex, is illustrated in Fig 1. Before the spousal loss we identified an overlap in males' overall, weekly expenditures between those who suffered bereavement and those who did not (difference = 4.46 Euros, 95% CI [-1.04, 9.95], t(175.03) = 1.60). This was not the case for bereaved females, for whom overall health care expenditures before spousal loss were lower

**Table 1. Distribution of socio-demographic variables in the study sample at inclusion.**

|  | Total | Males | Females |
|---|---|---|---|
| Sample Size | 924,958 | 413,163 (44.6%) | 511,795 (55.4%) |
| Age at Entry |  |  |  |
| 65–69 | 316,348 (34.2%) | 155,198 (37.6%) | 161,150 (31.5%) |
| 70–74 | 221,981 (24.0%) | 105,016 (25.4%) | 116,965 (22.9%) |
| 75–79 | 161,038 (17.4%) | 71,891 (17.4%) | 89,147 (17.4%) |
| 80–84 | 115,878 (12.5%) | 47,121 (11.4%) | 68,757 (13.4%) |
| 85plus | 109,713 (11.9%) | 33,937 (8.2%) | 75,776 (14.8%) |
| Affluence Index Group |  |  |  |
| 0–25 | 202,907 (21.9%) | 80,914 (19.6%) | 121,993 (23.8%) |
| 26–50 | 228,551 (24.7%) | 104,094 (25.2%) | 124,457 (24.3%) |
| 51–75 | 244,347 (26.4%) | 112,804 (27.3%) | 131,543 (25.7%) |
| 76–100 | 249,153 (26.9%) | 115,351 (27.9%) | 133,802 (26.1%) |
| Number of Children |  |  |  |
| Zero | 167,571 (18.1%) | 67,578 (16.4%) | 99,993 (19.5%) |
| One | 192,802 (20.8%) | 79,045 (19.1%) | 113,757 (22.2%) |
| Two | 335,897 (36.3%) | 157,763 (38.2%) | 178,134 (34.8%) |
| Three | 164,806 (17.8%) | 77,867 (18.8%) | 86,939 (17.0%) |
| Four or More | 63,882 (6.9%) | 30,910 (7.5%) | 32,972 (6.4%) |
| Number of Comorbidities |  |  |  |
| Zero | 325,027 (35.1%) | 147,858 (35.8%) | 177,169 (34.6%) |
| One | 228,207 (24.7%) | 100,301 (24.3%) | 127,906 (25.0%) |
| Two | 148,747 (16.1%) | 65,642 (15.9%) | 83,105 (16.2%) |
| Three | 95,980 (10.4%) | 42,637 (10.3%) | 53,343 (10.4%) |
| Four or More | 126,997 (13.7%) | 56,725 (13.7%) | 70,272 (13.7%) |

Data are n (%)

SD = Standard Deviation

than females who did not suffer bereavement (difference = -13.94 Euros, 95% CI [-18.11, -9.78], t(197.14) = -6.60), a pattern manifested also in all types of costs. The patterns of overall health care expenditures across bereavement stages, stratified on different age groups is plotted in Fig 2A & 2B for males and females respectively. First, we observe an overlap in the health care expenditures of bereaved and non-bereaved males in the period of two years before spousal loss in all age groups, except the 85-plus (difference = 16.24 Euros, 95% CI [4.59, 27.88], t (168.26) = 2.75). The pre-bereavement medical spending of females however exhibits a different behavior, with the gap in health care expenditures between the bereavement groups widening with older age. Second, there are incrementally higher health expenditures with age after spousal loss for both bereaved males and females when compared with the non-bereaved group.

The average effect of bereavement in health care expenditures was assessed with linear regression models, using a DiD approach. The results of the analysis, stratified on sex (also pooled) and age, are reported in Table 2. We calculated the gradual increase in health expenditures as the difference in differences coefficient one year before bereavement (DiD-pre) and one year after bereavement (DiD-post). Among males, one-year prior bereavement medical expenditures already increased 14 Euros per week (95% CI, 10 to 19) and one-after with another 42 Euros per week (95% CI, 36 to 48). Among women it increased with 5 Euros per week (95% CI, 0 to 9) before bereavement and 35 Euros per week (95% CI, 30 to 40) after

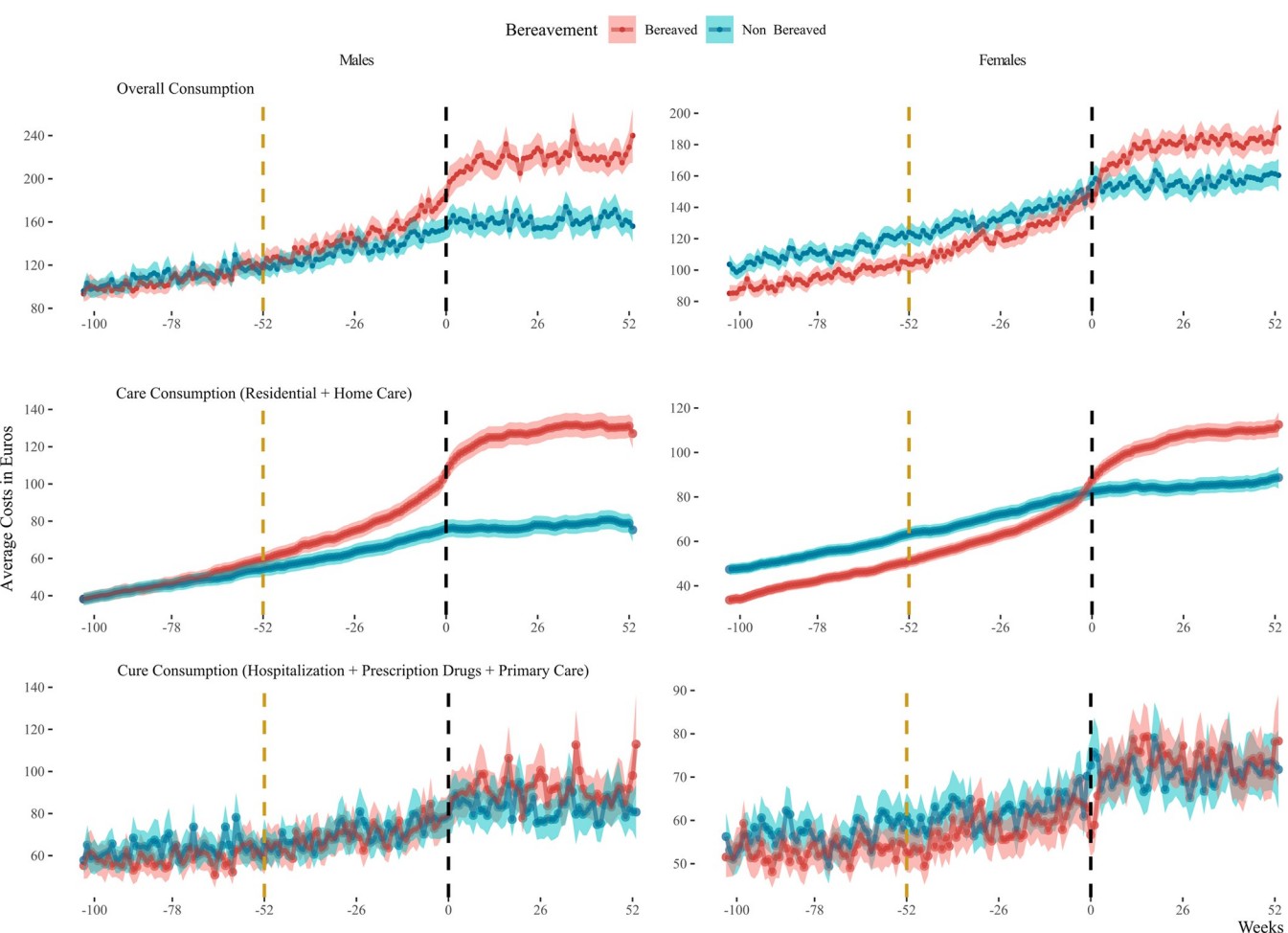

**Fig 1. Weekly average expenditures in Euros stratified on bereavement status and sex.** The red line represents individuals who suffered bereavement, the blue lines those who did not. Week 0 depicts the date of bereavement (black vertical line). Week -52 depicts the year before bereavement (yellow vertical line). The first column of the graph depicts the expenditures for males whereas the second one is for females. The first row illustrates the overall, aggregated health care expenditures of individuals (care- and cure- related), while the 2nd and 3rd row show the pattern specifically for care and cure expenditures, respectively.

bereavement. The higher increase in health care expenditures before and after bereavement in males compared to females was consistent for all age categories. The difference was markedly present at the oldest age categories, males ["85plus"], were spending approximately 37 Euros more per week [95% CI, 24 to 51] the year before bereavement when compared with two years before. Soon-to-be bereaved females did not manifest significant changes in their medical spending over a period of two years before bereavement. The R-squared of the models for the DiD analysis (-Pre and -Post) was 0.89 and 0.93 for males and females, respectively, being higher for older age groups where the relationship between time and expenditures was more linear. We observed sex differences in respect to spousal loss, with bereaved males spending 28 Euros [95% CI, 18 to 37] more per week compared with females, after adjusting for age, affluence index, number of comorbidities, and number of children. While the medical spending of the youngest ["65–69"] bereaved males did not significantly differ with that of females, the oldest ["85plus"] males exhibited an increased weekly spending of 59 Euros [95% CI, 33 to 84] compared with females of the same group.

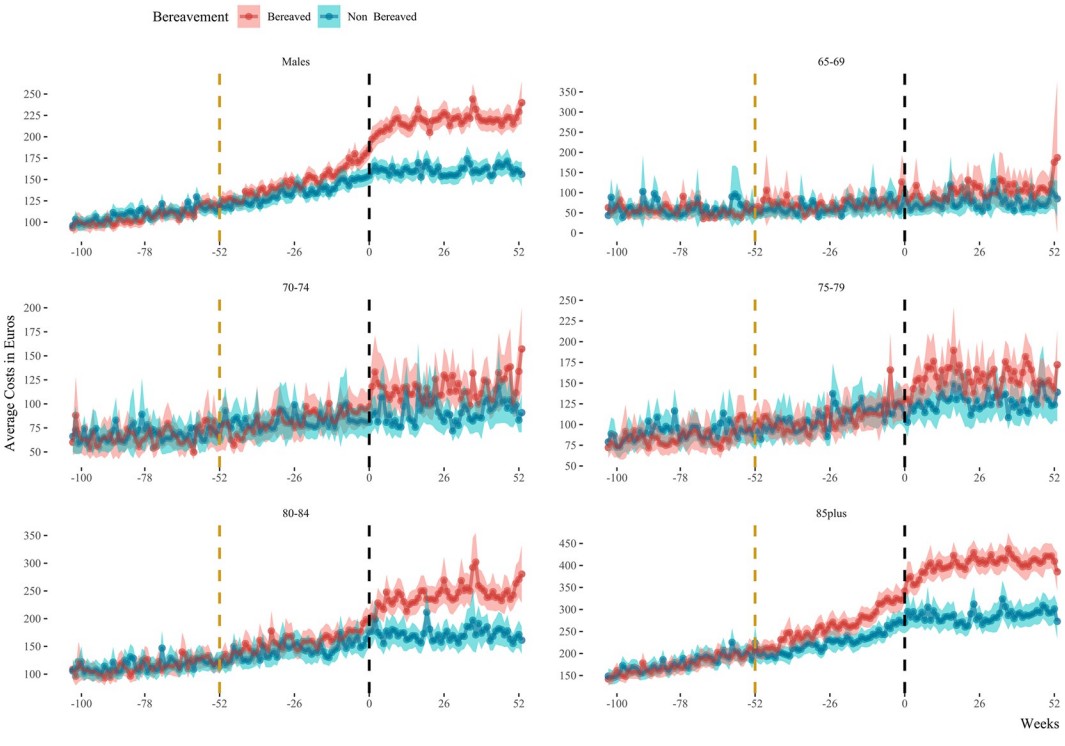

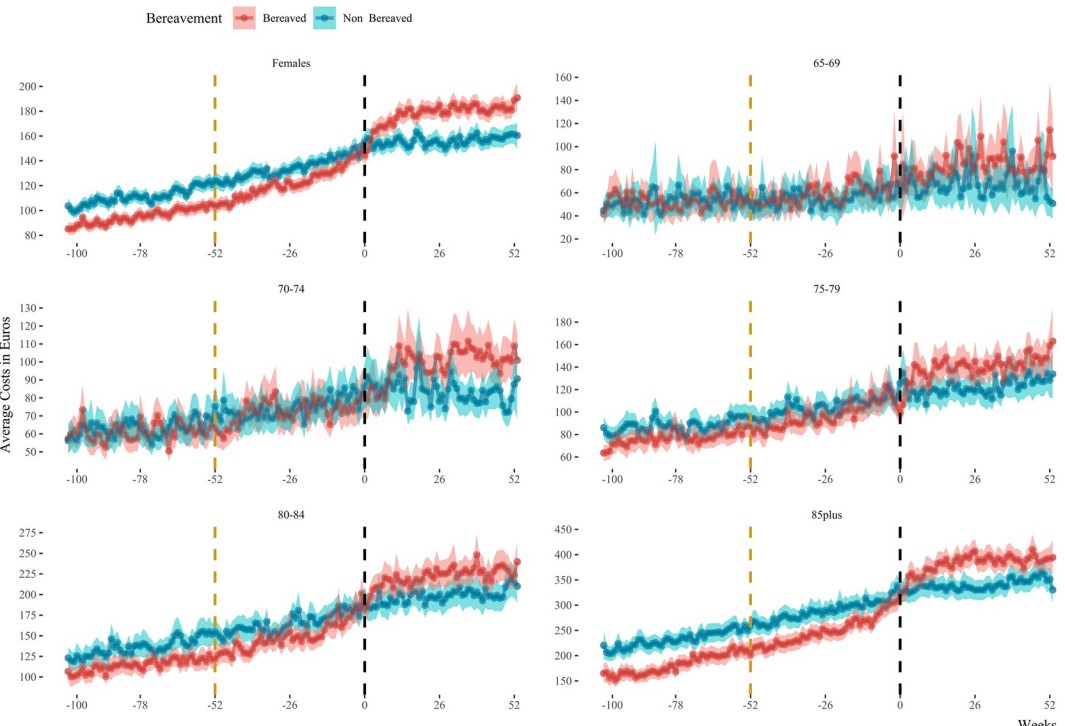

**Fig 2. a**. Overall average health care expenditures in Euros of males stratified on bereavement status and age. The red line represents individuals who suffered bereavement, the blue lines those who did not. Week 0 depicts the date of the standardized stressor of bereavement. Week -52 depicts the year before bereavement (yellow vertical line). The panel on the top left displays the time series of health care expenditures for males of all ages. The other panels show expenditures in various age strata. **b.** Overall average health care expenditures in Euros of females stratified on bereavement status and age. The red line represents

individuals who suffered bereavement, the blue lines those who did not. Week 0 depicts the date of the standardized stressor of bereavement. Week -52 depicts the year before bereavement (yellow vertical line). The panel on the top left displays the time series of health care expenditures for males of all ages. The other panels show expenditures in various age strata.

## Mortality hazard after spousal loss

First, we analyzed mortality in males and females of different ages for the first year after spousal loss. Table 3 depicts the results of the association of spousal bereavement with mortality, stratified on both sex and various age categories. We observed the widowhood effect to be the largest for the youngest age groups. While bereaved males are exhibiting higher mortality hazards in almost all age categories, bereaved females seem to have similar or even lower mortality hazard compared with non-bereaved ones, except for those in the '65–69' age category. When compared with the non-bereaved, bereaved males in the age group of '65–69' suffer a 70% higher mortality hazard the year after loss [HR 1.70, 95% confidence interval [CI], 1.40–2.08]. The corresponding hazard among females was found to be 27% higher [HR 1.27, 95% CI, 1.07–1.52].

Second, we illustrate the association of spousal bereavement with mortality hazard conditional on its duration in Fig 3. Mortality hazards were consistently lower in the first weeks after bereavement in both sexes of all age categories. This lower-than-expected mortality reversed into excess mortality, and it was highest in the first year since spousal loss, especially evident in the youngest age categories. Ultimately, bereaved males manifested higher excess in mortality hazards across the follow-up compared to non-bereaved ones, a pattern being more apparent for the younger age categories. We did not discern the same behavior in the females' bereavement groups, with the latter displaying similar or smaller hazards across time.

**Table 2. Difference in overall health care costs one-year before (DiD-Pre) and one-year after (DiD-Post) bereavement compared to those who did not.**

| Stratum | Age Category | DiD-Pre-Coefficient [95% CI] | DiD-Post Coefficient [95% CI] |
|---|---|---|---|
| Both Sexes (N = 38,027) | All | 9 [6, 13] | 38 [34, 43] |
| | 65–69 | 8 [4, 12] | 16 [10, 21] |
| | 70–74 | 1 [–2, 3] | 20 [17, 24] |
| | 75–79 | 4 [1, 8] | 27 [22, 31] |
| | 80–84 | 10 [5, 16] | 48 [41, 54] |
| | 85plus | 24 [13, 34] | 81 [70, 93] |
| Males (N = 12,935) | All | 14 [10, 19] | 42 [36, 48] |
| | 65–69 | 12 [4, 19] | 19 [10, 28] |
| | 70–74 | 2 [–3, 7] | 23 [18, 29] |
| | 75–79 | 6 [–1, 12] | 29 [22, 36] |
| | 80–84 | 15 [8, 23] | 56 [48, 65] |
| | 85plus | 37 [24, 51] | 82 [69, 95] |
| Females (N = 25,092) | All | 5 [0, 9] | 35 [30, 40] |
| | 65–69 | 5 [1, 9] | 13 [7, 18] |
| | 70–74 | 0 [–3, 2] | 17 [13, 21] |
| | 75–79 | 3 [–1, 8] | 24 [19, 29] |
| | 80–84 | 6 [–1, 12] | 39 [32, 46] |
| | 85plus | 10 [–1, 21] | 81 [71, 92] |

CI = Confidence Interval, DiD-Pre = Difference in differences coefficient for the period of two to one year pre-bereavement, DiD-Post = Difference in differences coefficient for the period of one year before to one year after bereavement. Coefficients are in values of Euros per week.

The models for all were adjusted for age.

The models for both sexes were further adjusted for sex.

**Table 3. Sex and age stratified hazard ratios (95% CI) for all-cause mortality of individuals who suffer bereavement in the first year after spousal loss compared to those who did not.**

| Stratum | Age Category | Hazard Ratio [95% CI] |
|---|---|---|
| Males (N = 413,163) | All | 1.13 [1.09, 1.18] |
| | 65–69 | 1.70 [1.40, 2.08] |
| | 70–74 | 1.32 [1.18, 1.48] |
| | 75–79 | 1.17 [1.06, 1.28] |
| | 80–84 | 1.14 [1.06, 1.23] |
| | 85plus | 0.97 [0.91, 1.04] |
| Females (N = 511,795) | All | 0.92 [0.88, 0.95] |
| | 65–69 | 1.27 [1.07, 1.52] |
| | 70–74 | 0.99 [0.90, 1.11] |
| | 75–79 | 0.88 [0.81, 0.96] |
| | 80–84 | 0.96 [0.89, 1.03] |
| | 85plus | 0.87 [0.82, 0.93] |

CI = Confidence Interval, models adjusted for Affluence index, number of children and number of comorbidities.

## Discussion

The current research analyzed the health care expenditures and mortality hazard of Danish individuals 65 years of age who experienced spousal bereavement. The main findings are as follows: 1. Females but not males had lower health care expenditures than non-bereaved females before experiencing bereavement; 2. Bereavement was associated with increase of health care expenditures and higher in males than females; 3. The oldest males were affected by the impending loss by showing signs of increasing health care consumption closer to the bereavement date; 4. The excess in mortality hazards after bereavement was evident in males, most prominent in young old, almost absent in the oldest old and most apparent in the first year after bereavement; 5. With the exception of the youngest ones, females exhibited signs of similar or smaller mortality hazards after bereavement; 6. The widowhood effect was persistent with signs of increasing trend even after a 6 year period after spousal loss in the youngest males.

The large sample size and extensive information of the Danish register data allowed us the unique opportunity to combine two different sources of data, health care spending and mortality, and investigate their association with spousal bereavement. The analysis of health care expenditures was insightful, allowing us to observe sex specific patterns in the behavior of individuals both before and after bereavement. In line with previous literature, we observed the effect of bereavement on health care costs being significant, increasing with age and higher for males [36, 37]. A previous Danish, register-based study on bereaved individuals showed no major sex differences in change in health care use, by analyzing health care use data related to prescriptions and primary care visits [25]. We note that this discrepancy could be due to the higher severity of males' diseases being reflected in increased, aggregated costs (covering both care and cure use) but not in the number of prescriptions and primary care visits, the latter being a less accurate measure of one's health status. Additionally, we enhance the strength of previous findings on the differential effect of bereavement on mortality by sex and age [19–21, 23, 24]. Novel is finding that the combined analyses show that bereavement had the greatest impact on males, especially in the youngest old.

Females who experienced bereavement had less health care expenditures already two years before spousal loss, a pattern we did not discern in males, who did not manifest substantial

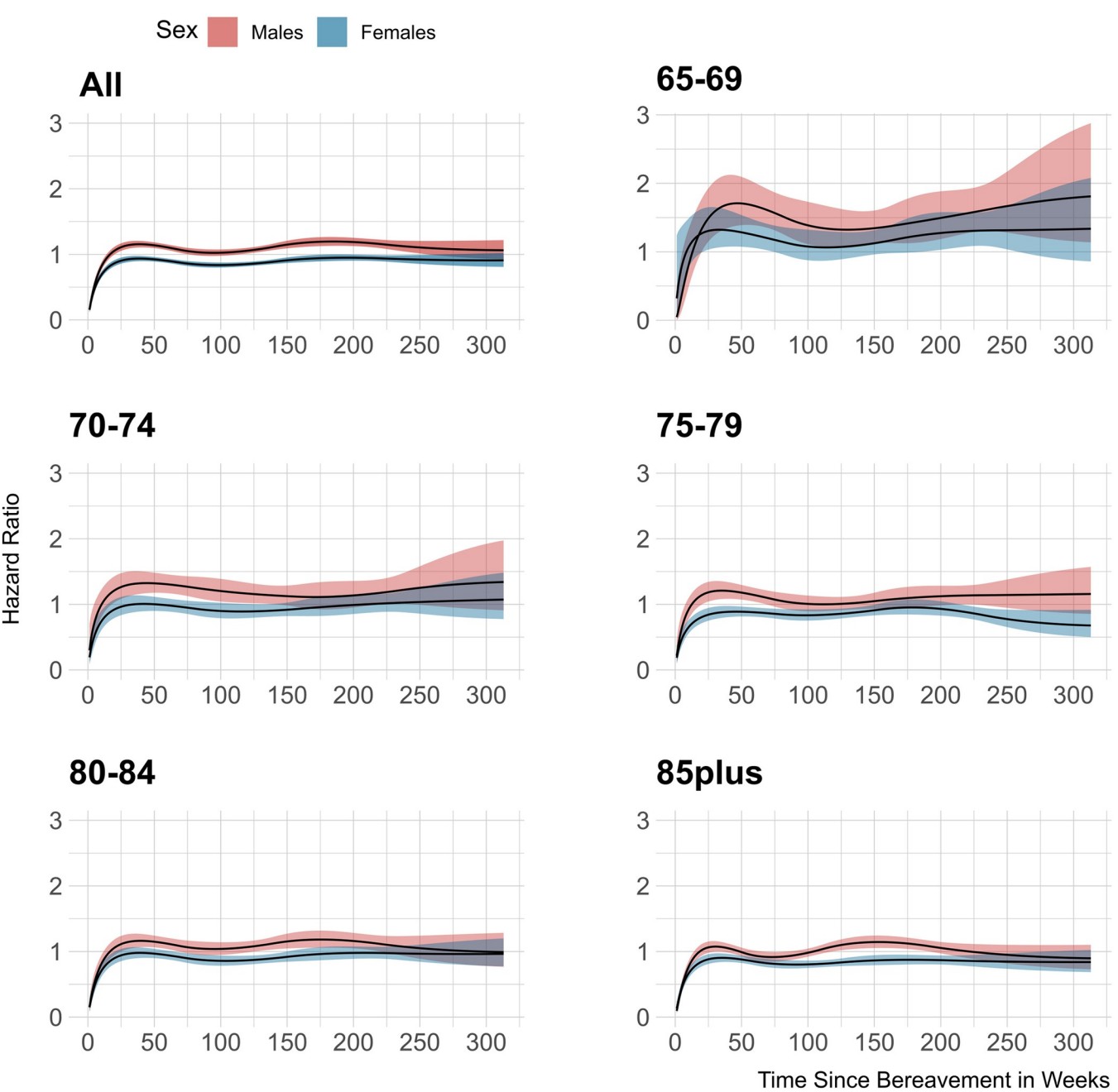

**Fig 3. Sex and age stratified hazard ratios for all-cause mortality based on bereavement status.** The panel on the top-left shows the overall hazard ratio for mortality as a function of time since bereavement for males and females respectively. The other panels show hazard ratios in various age strata.

differences in their pre-bereavement status, whereas the oldest ["85plus"] showed an anticipatory increase in expenditures. The phenomenon of lower pre-bereavement medical spending for females was observed for those over the age of 75 and not in the youngest ones. The aforementioned can be attributed to the scenario that older soon-to-be bereaved females tend to ignore their own health status in order to provide caregiving to their spouse. Conversely, the inability of soon-to-be deceased females to provide caregiving to their soon-to-be bereaved spouse, might be explanatory of the anticipatory increase in spending of the latter.

The sex differences in health expenditures disappears after bereavement, with their respective health care expenditures climbing up to levels higher to the non-bereaved individuals. Ultimately, males were more in need for health care services after spousal loss, indicating a higher vulnerability, a finding also supported by the higher excess of mortality hazards compared with females. In addition, medical spending of the oldest bereaved males was substantially higher than their reference group even before bereavement. With regards to the type of health care costs, those related to care were less variable than the cure ones, which can be explained by the intrinsic nature and design of long-term care for the former category. Ultimately, care consumption for old, bereaved males was substantially higher than the reference group even before bereavement. We hypothesize that this can be explained by the inability of males to maintain their functional status, and provide adequate spousal caregiving and housekeeping while their wife is ill and soon-to-be deceased.

In line with the literature [20, 23, 24], we found bereavement to increase mortality hazard of individuals over the age of 65. However, we observed that depending on the sex and age of the cohort participants, the effect is rather different. Specifically, a different effect of bereavement in males and females was discerned, with males being the ones susceptible to bereavement by experiencing excessive mortality hazards, although decreasing with age. On the contrary, with the exception of the youngest old, bereaved females exhibited similar or smaller mortality hazards compared with the non-bereaved ones, a pattern being more evident with older age. The aforementioned pattern can possibly be attributed to the caregiving behavior of older females to their male sick spouses ceasing after bereavement which might be a beneficial, in terms of health status, factor. Moreover, we consider the asymmetry of the observed association between the two sexes as an argument against the phenomenon of shared confounding, since if there was an unobserved cause associated with bereavement and mortality, we would expect the direction to remain unchanged between males and females.

We were able to extract insights regarding the dynamics of resilience for both sexes and for different age groups in response to bereavement, by computing the hazard ratios as a function of time since spousal loss. An observed hazard ratio of 1 was deemed as an equilibrium state in which bereaved individuals stabilize in the same mortality hazard as non-bereaved ones, translating to adaptivity or recovery from the stressor of bereavement. Following that logic, bereaved females, except the youngest ones, exhibited robust signs of resilience, with no significant adverse dispersion from their initial equilibrium state, manifesting similar or smaller mortality hazards compared with non-bereaved ones. On the contrary, bereaved males did not show clear signs of returning to the equilibrium state, but rather displayed mortality hazards which rose sharply soon after bereavement and then slowly diminishing, but mostly remaining higher than their respective control group. Therefore, bereavement seems to shift males into a prolonged allostatic response, away from their initial state of functionality, increasing their health vulnerability without signs of bouncing back. That pattern was most evident in the youngest bereaved males, for whom the widowhood effect was persistent and increasing even after the first three years since spousal loss. Opposite to males, females exhibited a proper allostatic response, initiating after spousal bereavement, sustained for a short time interval, with mortality hazards eventually normalizing. The aforementioned insights are further supported by previous scientific work, documenting sex and age-related differences in resilience and vulnerability of stressors across life [12].

Ultimately, there was a pattern of higher utilization of health care services in the older age groups which seems to be both needed and beneficial for both sexes. We attribute that finding on the fact that the baseline mortality hazard is high for the oldest groups, which are frailer, more likely to have multiple comorbidities, and experiencing difficulties with housekeeping. We hypothesize that one of the main reasons for observing similar but not higher mortality

hazards between the oldest bereaved and non-bereaved individuals, is the high utilization of cure and especially care services after spousal loss. Conversely, the youngest bereaved persons, both males and females, had the highest excess in mortality hazards, while also exhibiting the smallest increase in medical spending after the loss. The aforestated finding was expected given that bereavement in the younger age appears to be more of an unexpected health shock rather than a natural consequence of frailty due to ageing. Moreover, the smaller medical spending, compared with the oldest old, can be explained by the smaller baseline mortality hazard of the younger group. However, we found evidence that the widowhood effect in the young individuals might be prominent in the first year after loss, but also persistent in the long term. Thereby, we propose the development and maintenance of bereavement support and palliative care services with immediate access for young, bereaved individuals in order to reduce the excess hazards which seem to be present not only in the first year after spousal loss but even in the long-term.

The strength of the present investigation is the ability to study the association of bereavement and mortality hazard in a sample of approximately 925,000 Danish individuals, a size which to our knowledge is one of the biggest in bereavement research. The extensive information available in the Danish national registers, allowed us to investigate a plethora of health care expenditure types, ranging from care to cure related ones, with that being a strength of the study enhancing the understanding of how spousal bereavement affects the health of individuals. The flexible, using splines, modelling of the hazard ratios between bereaved and non-bereaved individuals provided a clearer overview of bereavement's effect across time, hence we also consider it as a strength. However, our study has its limitations. First, the results are not representative of the whole Denmark's population since the data include only older adults who have experienced spousal bereavement. Nevertheless, we assume that spousal loss in younger individuals (less than 65)–although relatively seldom—might have an effect similar to our cohort's youngest participants and possibly worse. Second, the results might not generalize to countries with different health care systems and strategies, which might alter the results of the expenditures analysis. Last, we note that the estimation of the direct effect of spousal bereavement on mortality is not plausible, since there will always be common unobserved causes of similarities (assortative mating) in mortality risk between spouses, that no exhaustive amount of data can solve. Thereby, we refrain from interpreting our results from a pure causal perspective.

## Supporting information

**S1 Table. Description of the two study designs investigating the association of spousal loss with medical spending and mortality.**
(DOCX)

**S2 Table. STROBE checklist for cohort studies.**
(DOCX)

## Author Contributions

**Conceptualization:** Alexandros Katsiferis, Rudi G. J. Westendorp.

**Data curation:** Laust Hvas Mortensen.

**Formal analysis:** Alexandros Katsiferis.

**Funding acquisition:** Rudi G. J. Westendorp.

**Methodology:** Alexandros Katsiferis, Samir Bhatt, Swapnil Mishra, Rudi G. J. Westendorp.

**Software:** Alexandros Katsiferis.

**Writing – original draft:** Alexandros Katsiferis, Samir Bhatt, Rudi G. J. Westendorp.

**Writing – review & editing:** Alexandros Katsiferis, Laust Hvas Mortensen, Swapnil Mishra, Rudi G. J. Westendorp.

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
