## [Decision Letter · Decision Letter 0]

31 Jan 2023

PONE-D-22-23580Sex Dimorphism in Health Expenditures and Mortality after Spousal Bereavement: A retrospective, register-based Danish cohort studyPLOS ONE

Dear Dr.
Katsiferis,

Thank you for submitting your manuscript to PLOS ONE. After careful consideration, we feel that it has merit but does not fully meet PLOS ONE’s publication criteria as it currently stands. Therefore, we invite you to submit a revised version of the manuscript that addresses the points raised during the review process. Given the extensive comments by the reviewer, I am returning your manuscript for revision.  Once completed the revised manuscript will be sent out to additional reviewers.

We look forward to receiving your revised manuscript.

Kind regards,

Rosemary Frey

Academic Editor

PLOS ONE

Journal Requirements:

“This work is supported by The Novo Nordisk Foundation (https://novonordiskfonden.dk/en/) Challenge Programme for the project Harnessing the Power of Big Data to Address the Societal Challenge of Aging (NNF17OC0027812). The funders had no role in study design, data collection and analysis, decision to publish, or preparation of the manuscript.”

“I have read the journal's policy and the authors of this manuscript have the following competing interests: LHM is employed at Statistics Denmark, the national Danish Statistics office. This does not alter our adherence to PLOS ONE policies on sharing data and materials.”

We note that one or more of the authors are employed by a commercial company: Statistics Denmark

Reviewers' comments:

Reviewer's Responses to Questions

**Comments to the Author**

1. Is the manuscript technically sound, and do the data support the conclusions?

Reviewer #1: Yes

2. Has the statistical analysis been performed appropriately and rigorously? 

Reviewer #1: Yes

3. Have the authors made all data underlying the findings in their manuscript fully available?

Reviewer #1: No

4. Is the manuscript presented in an intelligible fashion and written in standard English?

Reviewer #1: Yes

5. Review Comments to the Author

Reviewer #1: this is an interesting paper on the association between spousal bereavement and Health Expenditures or Mortality and modifications by age and sex with a special focus on sex differences.

Sex differences in mortality in favour of women are well documented in the literature, whereas the finding of increased healthcare expenses is more novel as is also the increased healthcare expenses of pre bereaved women.

Title:

I suggest changing “Sex Dimorphism” to “Sex differences” as “Dimorphism” denotes a trait that occurs in two distinct forms or morphs within a given species and thus leads the thoughts on causal biological effects on the differences which cannot be inferred based on the results (and also change it throughout the text). I also suggest omitting “retrospective” as all cohort studies, in principle, are prospective, and given that it is a Danish registry-based study, the data were collected prospectively. The analysis in cohort studies is always done retrospectively.

I guess it is health care Expenditures which is meant as health is no monetary currency.

Suggested title: Sex differences in Health Care Expenditures and Mortality after Spousal

Bereavement: A register-based Danish cohort study

Introduction:

In Oksuzyan et al 2011 no major sex differences in change in health care use (hospitalisations and medicine use (prescriptions) were found, which tends to contradict the findings in the presented study. The study should be mentioned together with the discrepancy, and also acknowledge the findings in the study of an age effect and sex effect (e.g. table 2 in Oksuzyan et al 2011). The discrepancy could e.g., be due to the higher severity of men's diseases being reflected in increased cost (total health care use measured as services codes, i.e., better measure) but not in the number of prescriptions (less accurate measure).

Methods:

To be able to replicate the study, a more precise description of the used registries should be made. I guess that “Sygesikringsregisteret (SSR)” was used for health care expenditures. The Danish Health Data Authority holds the registry, and Statistics Denmark holds a copy. Thus, a reference to a paper describing the registry at The Danish Health Data Authority should be made (e.g. Scandinavian Journal of Public Health, 2011; 39(Suppl 7): 34–37, or similar). Similarly should be done for the rest of the used registries.

How were comorbidities measured? Hospitalisations? medication use? Please give a more precise description.

The spline-based survival analysis and best-fit model approach make the results for mortality trustworthy. I wonder if it is possible to address how well the difference in differences linear regression model fitted the observed data?

It was not possible for me to see the “S1 Table”?

Results

The results are well-presented and easy to follow.

If possible, confidence should be added for each line in figure 1 and figure 2.

It is somewhat unclear if sex differences exist between bereaved women and men across ages.

An additional table with differences in differences for post bereavement for the two sexes would make this easier for the reader to understand the results on sex differences (which is in focus given the title).

Discussion

The discussion is well written and gives a comprehensive discussion with suggestions for the observed results.

The hypotheses on reasons for the sex differences and age differences would, however, be more valid given that previous studies that had indicated similar effects were referenced. E.g., when mentioning Vulnerability and Resilience, a paper like Hodes et al. could beneficially be mentioned and referenced to support the hypothesis given (e.g., L384).

That bereaved women had a lower health care expenditure before bereavement than other women could, as suggested, be attributed to the scenario that older soon-to-be bereaved females tend to ignore their health status to provide caregiving to their spouse. But why is a similar scenario not seen in men? E.g., a hypothesis is that: men do not lose focus on their health when their spouse is sick, while the opposite is true for women.

One minor comment:

L322: ” Females but not males had lower healthcare expenditures before experiencing 32bereavement” should be changed to: ”Females but not males had lower healthcare expenditures than not bereaved women before experiencing bereavement”

References:

Oksuzyan A, Jacobsen R, Glaser K, Tomassini C, Vaupel JW, Christensen K. Sex Differences in Medication and Primary Healthcare Use before and after Spousal Bereavement at Older Ages in Denmark: Nationwide Register Study of over 6000 Bereavements. J Aging Res. 2011;2011:678289. doi:10.4061/2011/678289

Hodes GE, Epperson CN. Sex Differences in Vulnerability and Resilience to Stress Across the Life Span. Biol Psychiatry. 2019 Sep 15;86(6):421-432. doi: 10.1016/j.biopsych.2019.04.028. Epub 2019 May 7. PMID: 31221426; PMCID: PMC8630768.

6. PLOS authors have the option to publish the peer review history of their article (what does this mean?). If published, this will include your full peer review and any attached files.

Reviewer #1: **Yes: **Rune Lindahl-Jacobsen

---

## [Author Response · Author response to Decision Letter 0]

16 Feb 2023

Our full response to the reviewer is attached as a separate Word file.

---

## [Decision Letter · Decision Letter 1]

27 Feb 2023

Sex Differences in Health Care Expenditures and Mortality after Spousal Bereavement: A register-based Danish cohort study

PONE-D-22-23580R1

Dear Dr. Katsiferis,

We’re pleased to inform you that your manuscript has been judged scientifically suitable for publication and will be formally accepted for publication once it meets all outstanding technical requirements.

Kind regards,

Rosemary Frey

Academic Editor

PLOS ONE

Additional Editor Comments (optional):

Reviewers' comments:

Reviewer's Responses to Questions

**Comments to the Author**

1. If the authors have adequately addressed your comments raised in a previous round of review and you feel that this manuscript is now acceptable for publication, you may indicate that here to bypass the “Comments to the Author” section, enter your conflict of interest statement in the “Confidential to Editor” section, and submit your "Accept" recommendation.

Reviewer #1: All comments have been addressed

2. Is the manuscript technically sound, and do the data support the conclusions?

Reviewer #1: Yes

3. Has the statistical analysis been performed appropriately and rigorously? 

Reviewer #1: Yes

4. Have the authors made all data underlying the findings in their manuscript fully available?

Reviewer #1: Yes

5. Is the manuscript presented in an intelligible fashion and written in standard English?

Reviewer #1: Yes

6. Review Comments to the Author

Reviewer #1: The Authors have addressed all comments satisfactorily and the manuscript has improved. The paper brings new insight into sex differences in the economic consequences of major life events. I have no further comments.

7. PLOS authors have the option to publish the peer review history of their article (what does this mean?). If published, this will include your full peer review and any attached files.

Reviewer #1: **Yes: **Rune Lindahl-Jacobsen

---

## [Editor Report · Acceptance letter]

1 Mar 2023

PONE-D-22-23580R1 

Sex Differences in Health Care Expenditures and Mortality after Spousal Bereavement: A register-based Danish cohort study 

Dear Dr. Katsiferis:

I'm pleased to inform you that your manuscript has been deemed suitable for publication in PLOS ONE. Congratulations! Your manuscript is now with our production department. 

Kind regards, 

on behalf of

Dr. Rosemary Frey 

Academic Editor

PLOS ONE